# Use of Trifluoro-Acetate Derivatives for GC-MS and GC-MS/MS Quantification of Trace Amounts of Stera-3β,5α,6β-Triols (Tracers of Δ^5^-Sterol Autoxidation) in Environmental Samples

**DOI:** 10.3390/molecules28041547

**Published:** 2023-02-06

**Authors:** Claude Aubert, Jean-François Rontani

**Affiliations:** 1Laboratoire de Pharmacocinétique et Toxicocinétique (Equipe Associée 3286), Faculté de Pharmacie, 13385 Marseille, France; 2CNRS/IRD, Mediterranean Institute of Oceanography (MIO), UM 110, Aix-Marseille University, Université de Toulon, 13288 Marseille, France

**Keywords:** stera-3β,5α,6β-triols, autoxidation tracers, derivatization, trifluoroacetylation, validation, GC-EI(QTOF), GC-MS/MS, environmental samples

## Abstract

Stera-3β,5α,6β-triols make useful tracers of the autoxidation of Δ^5^-sterols. These compounds are generally analyzed using gas chromatography–mass spectrometry (GC-MS) after silylation. Unfortunately, the 5α hydroxyl groups of these compounds, which are not derivatized by conventional silylation reagents, substantially alter the chromatographic properties of these derivatives, thus ruling out firm quantification of trace amounts. In this work, we developed a derivatization method (trifluoroacetylation) that enables derivatization of the three hydroxyl groups of 3β,5α,6β-steratriols. The derivatives thus formed present several advantages over silyl ethers: (i) better stability, (ii) shorter retention times, (iii) better chromatographic properties and (iv) mass spectra featuring specific ions or transitions that enable very low limits of detection in selected ion monitoring (SIM) and multiple reaction monitoring (MRM) modes. This method, validated with cholesta-3β,5α,6β-triol, was applied to several environmental samples (desert dusts, marine sediments and particulate matter) and was able to quantify trace amounts of 3β,5α,6β-steratriols corresponding to several sterols: not only classical monounsaturated sterols (e.g., cholesterol, campesterol and sitosterol) but also, and for the first time, di-unsaturated sterols (e.g., stigmasterol, dehydrocholesterol and brassicasterol).

## 1. Introduction

Autoxidation (free radical oxidation) of Δ^5^-sterols mainly affords 7α- and 7β-hydroperoxides and, to a lesser extent, 5α/β,6α/β-epoxysterols and 3β,5α,6β-trihydroxysterols [1]. The 7α- and 7β-hydroperoxides have been ruled out as possible markers of autoxidation processes in the environment due to their instability (fast degradation under environmental conditions) and lack of specificity (formation is also possible by allylic rearrangement of photochemically produced 5α-hydroperoxysterols) [2,3]. Unfortunately, the highly specific 5α/β,6α/β-epoxysterols have also been ruled out as they are too unstable under environmental conditions; they are quickly hydrolyzed to their corresponding triols by epoxide hydrolase [4] and under acidic conditions [5]. The 3β,5α,6β-trihydroxysterols, which are stable and only produced during autoxidation processes, have thus been proposed as specific tracers of sterol autoxidation in the environment [2,3].

Electron ionization (EI) provides more structural information than the soft ionization techniques such as electrospray ionization (ESI) or atmospheric pressure chemical ionization (APCI) employed in HPLC-MS analyses [6], and so quantification of Δ^5^-sterols and their oxidation products in environmental samples is most often performed using gas chromatography–electron ionization mass spectrometry (GC-EIMS). GC-EIMS analyses are generally carried out on a nonpolar silicone stationary phase after silylation [2,5,7,8,9]. Silylation of sterol involves the replacement of the hydrogen of the hydroxyl group with an alkylsilyl (often trimethylsilyl) group. Trimethylsilyl (TMS) derivatives are highly volatile, thermally stable and present outstanding gas chromatographic characteristics. Moreover, the EI mass spectra of these derivatives often exhibit a significant [M-15]^+^ ion formed by the loss of a silicon-bonded methyl group, which is very useful for determining molecular mass, and are also very informative for structural elucidations [10,11]. However, since TMS derivatives can lose easily trimethylsilanol molecules under the effect of moisture, a short delay between derivatization and injection is needed. Despite this drawback, silylation is a very popular derivatization method and is often employed during sterol quantification using GC-MS [12,13,14,15,16]. Unfortunately, steric hindrance makes complete silylation of 3β,5α,6β-trihydroxysterols difficult, and common silylation reagents (such as bis(trimethylsilyl)trifluoroacetamide (BSTFA)/pyridine) afford derivatives that are only silylated at C-3 and C-6 [17]. Trisilylated derivatives can be obtained after treatment with BSTFA/dimethylsulfoxide (DMSO) [18], but conversion is still not complete (yields are close to 50%) and this treatment is still too complex to be applied for analysis of trace amounts in environmental samples. It is absolutely necessary to eliminate the DMSO before injection into the chromatographic column, and this operation (requiring: (i) addition of water, (ii) extraction with solvents and (iii) resilylation of 3β and 6β hydroxyl groups) cannot be carried out without significant losses of the lipidic material. The presence of a polar nonderivatized hydroxyl group at C-5 in the disilylated derivative strongly alters its chromatographic characteristics and leads to the formation of tailing peaks that substantially limit the sensitivity of the analyses [19].

Acetylation of steroids with acetic anhydride and trifluoroacetic anhydride is also very common [20,21,22,23]. However, it is generally considered that trimethylsilyl derivatives are more suitable for the GC-MS characterization and quantitation of sterols than acetate derivatives [16]. Acetylation involves replacement of the mobile hydrogen atoms of the hydroxyl groups of sterols with acyl or trifluoro acyl groups. Halogenated acyl groups enhance the electron affinity of the derivative and produce very specific fragmentation patterns in mass spectrometry [24]. It should be noted, however, that the use of fluorinated anhydrides requires the removal of any excess or byproducts prior to GC analysis to prevent deterioration of the column [25]. To our knowledge, trifluoroacetylation has not been used in the case of 3β,5α,6β-steratriols. 

In this work, we set out to develop a trifluoroacetylation method able to derivatize the three hydroxyl groups of 3β,5α,6β-trihydroxysterols in order to reduce analyte adsorption in the GC system and improve detector response, peak separation and peak symmetry. We used trifluoroacetic anhydride, which is well known to be highly reactive in the case of steric hindrance [25]. This derivatization technique was then validated using environmental samples (desert dusts, marine sediments and particulate matter), where it allowed the detection of traces of several triols resulting from the oxidation of mono- and di-unsaturated sterols.

## 2. Results and Discussion

### 2.1. Formation and Characterization of Trifluoroacetate Derivative of Cholesta-3β,5α,6β-Triol

Reaction of cholesta-3β,5α,6β-triol with trifluoroacetic anhydride in tetrahydrofurane (THF) under the conditions described in Section 3.2 afforded a trifluoroacetate derivative at high yield (>95%). As expected, this derivative presented better chromatographic characteristics (shorter retention time and better peak shape) than the corresponding *bis*-trimethylsilyl ether (Figure 1). It is well known that the introduction of fluorine atoms strongly enhances analyte volatility and thus reduces analyte retention time [26]. Due to its high content of fluorine atoms (nine per molecule), the trifluoroacetate derivative of cholesta-3β,5α,6β-triol eluted 9 min faster than the corresponding disilylated derivative (Figure 1) and 1.5 min faster than cholesterol trifluoroacetate on the 30 m capillary column employed. 

Although negative inductive effects of the fluorine atoms in a derivatized product may drive hydrolysis in the presence of moisture [27], here, the trifluoroacetate derivative of cholesta-3β,5α,6β-triol was found to be highly stable. Indeed, in contrast to the corresponding TMS derivative, which was hydrolyzed in a few days, it could be stored at 4 °C for several months without significant alteration.

The EI(TOF) mass spectrum of the cholesta-3β,5α,6β-triol trifluoroacetate derivative (Figure 2A) exhibited ions at *m/z* 594.3134 (**b^+•^**), 480.3209 (**c^+•^**) and 366.3273 (**d^+•^**) corresponding to the successive loss of one, two and three neutral molecules of trifluoroacetic acid by the molecular ion (**a^+•^**), respectively (Figure 3). Note that the abundance of the **c^+•^** ion resulted from the formation of a stable conjugated enol ester group. An ion at *m/z* 367.1876 (**e^+^**) resulting from the loss of two molecules of trifluoroacetic acid and the side-chain was also formed. The shift of ions **b^+•^**, **c^+•^** and **d^+•^** by 7 *m/z* units and the lack of shift of the **e^+^** ion observed in the EI(TOF) mass spectrum of cholest-5-en-25,26,26,26,27,27,27-d_7_-3β,5α,6β-triol trifluoroacetate derivative (Figure 2B) further supports these attributions. Unfortunately, due to its instability under electron impact, the molecular peak of the cholesta-3β,5α,6β-triol trifluoroacetate derivative was not observable in its EI(TOF) mass spectrum. We therefore used electron-capture negative ionization (ECNI), which is generally considered a soft ionization technique that yields a mass spectral pattern with less fragmentation than EI ionization [28]. The ECNI mass spectrum of the cholesta-3β,5α,6β-triol trifluoroacetate derivative (Figure 2C) appeared to be dominated by a peak at *m/z* 113 corresponding to the anion CF_3_-COO^−^ and a smaller molecular peak at *m/z* 708, attesting that the observed derivative was well triacetylated.

Based on its abundance (Figure 2A) and specificity, the **c^+•^** ion corresponding to [M—2CF_3_COOH]^+•^ was selected as the target ion for selected ion monitoring (SIM)-based quantification of the main 3β,5α,6β-steratriol trifluoroacetate derivatives present in environmental samples. Due to its high specificity, the less abundant **b^+•^** ion corresponding to [M—CF_3_COOH]^+•^ constituted a useful qualifier allowing confirmation of the identifications. Collision-induced dissociation (CID) analyses (Figure 4) allowed selection of the efficient transition **c^+•^** → **f^+^** [M—3CF_3_COOH—CH_3_]^+^ corresponding to the loss of a neutral molecule of CF_3_COOH and a methyl radical by the **c^+•^** ion (Figure 3) for multiple reaction monitoring (MRM) analyses (Table 1).

### 2.2. Validation of the Derivatization Method

Validation of the derivatization method was carried out using the cholesta-3β,5α,6β-triol trifluoroacetate derivative. Results of linearity tests in SIM and MRM modes are presented in Table 2. In the concentration range tested here (0.2325–46.5 ng/mL), the coefficients of determination of the linear regression curves were better than 0.995 and the intercepts did not differ significantly from 0.

Table 3 reports the reproducibility of this derivatization technique. The precision (given by the standard deviation) and accuracy (defined as the difference between obtained concentration and expected concentration) were acceptable over the concentration range.

The limit of detection (LOD) (defined by a signal-to-noise ratio of 5) was about 25.8 and 0.78 pg injected in SIM and MRM modes, respectively. For comparison, the LOD obtained for the corresponding disilylated derivative was 0.62 ng in SIM mode with the target ion *m/z* 456 corresponding to [M—TMSOH—H_2_O]^+•^. Due to the higher specificity of their MRM transitions, trifluoroacetate derivatives are much more suitable for the analysis of trace amounts of 3β,5α,6β-steratriols in complex samples than silylated derivatives.

### 2.3. Application to Different Environmental Samples

In an application of the derivatization method, 3β,5α,6β-steratriol trifluoroacetate derivatives originating from the autoxidation of common Δ^5^-sterols were quantified in total lipid extracts (TLEs) of several environmental samples (desert dusts, marine sediments and particulate matter). The results obtained are summarized in Table 4 and Table 5. This method allowed precise quantification of 3β,5α,6β-triols derived from classical monounsaturated Δ^5^-sterols: cholest-5-en-3β-ol (cholesterol), 24-methylcholest-5-en-3β-ol (campesterol) and 24-ethylcholest-5-en-3β-ol (sitosterol) (Table 4, Figure 5). The double peak observed in the case of the transition *m/z* 494 → *m/z* 365 (Figure 5) resulted from the well-known production of a mixture of 24-methylcholesterol epimers by eukaryotic organisms, wherein campesterol (24(*R*)-methylcholest-5-en-3β-ol) and dihydrobrassicasterol (24(*S*)-methylcholest-5-en-3β-ol) are found in variable proportions [29,30]. Due to their excellent chromatographic properties, diastereoisomeric 3β,5α,6β-steratriol trifluoroacetate derivatives could be easily separated, while the corresponding disilylated derivatives coeluted and showed only a small tailing peak.

Interestingly, we also detected unsaturated triols deriving from 5,22-di-unsaturated sterols, i.e., cholesta-5,22(*E*)-dien-3β-ol (dehydrocholesterol), 24-methylcholesta-5,22(*E*)-dien-3β-ol (brassicasterol) and 24-ethylcholesta-5,22(*E*)-dien-3β-ol (stigmasterol) (Table 5, Figure 6), which have never previously been described in the literature. Note that the mass spectra of these derivatives (Figure 7) were similar to the mass spectra of monounsaturated sterols (Figure 2A).

Sterols are commonly used as tracers for specific classes of organisms in environmental samples [31,32,33]. Due to their relative stability, these compounds and their degradation products are also excellent biomarkers for tracing the diagenetic transformation of organic matter [2,3,34,35,36]. The quantification of 3β,5α,6β-steratriols (autoxidation tracers) [2,3] and their corresponding sterols in natural samples thus provides valuable information on the oxidation state (and thus alteration) of specific organisms (e.g., higher plants, seagrasses, phytoplankton, zooplankton and fungi). For example, the presence of the triol corresponding to brassicasterol (a sterol present in many species of phytoplankton [33,35]) in the sediments of the Baffin Sea (Table 5) attests to the intervention of autoxidative processes in these organisms. On the other hand, the detection of high proportions of the triol corresponding to sitosterol (a major sterol of higher plants [33,35]) in the analyzed samples (Table 4) is indicative of the presence of strongly oxidized higher-plant material. Due to the high specificity of some sterols [33,35], the corresponding 3β,5α,6β-steratriols could even give indications concerning the autoxidation of specific phyla of phytoplankton (e.g., diatoms, prymnesiophytes and chlorophytes).

## 3. Materials and Methods

### 3.1. Chemicals

Trifluoroacetic anhydride (TFAA), cholesta-3β,5α,6β-triol, sterols, cholest-5-en-25,26,26,26,27,27,27-d_7_-3β-ol, *meta*-chloroperoxybenzoic acid, H_2_O_2_, N,O-bis(triméthylsilyl)trifluoroacetamide (BSTFA) and chemical reagents were obtained from Sigma-Aldrich. The synthesis of standards of the stera-3β,5α,6β-triols corresponding to campesterol, sitosterol, brassicasterol, dehydrocholesterol and stigmasterol involved epoxidation with *meta*-chloroperoxybenzoic acid in dry methylene chloride, and subsequent acid-catalyzed hydrolysis [37]. These compounds were purified subsequently using column chromatography as described below for the internal standard.

The internal standard used (cholesta-25,26,26,26,27,27,27-d_7_-3β,5α,6β-triol) was synthesized by KI/H_2_O_2_ oxidation of the corresponding heptadeuterosterol [38]. Cholest-5-en-25,26,26,26,27,27,27-d_7_-3β-ol (5 mg), KI (2.2 mg) and dioxane/water (0.9 mL, 2:1, *v*/*v*) were placed in a 20 mL flask, and then H_2_SO_4_ (98%, 5 µL) and H_2_O_2_ (30%, 10 µL) were added sequentially at room temperature under magnetic stirring. After stirring for 1 h at room temperature, the system was stirred for 3 h at 60 °C and the reaction mixture was then neutralized with anhydrous Na_2_CO_3_ (2.2 mg) and treated with a saturated solution of Na_2_SO_3_ (4 mL). The crude triol was extracted twice (4 mL) with ethyl acetate and the organic extracts were evaporated to dryness under nitrogen at 50 °C. The crude triol was then purified using column chromatography (silica, Kieselgel 60 with 55% water, 6 × 0.6 cm). The column was conditioned with CH_2_Cl_2_. After elimination of the residual sterol with CH_2_Cl_2_ (8 mL), the triol was eluted with CH_3_CN (6 mL).

The standard solutions of cholesta-3β,5α,6β-triol and internal standard were prepared by dissolving 10 mg measures of these compounds (weighted) in 10 mL of methanol. Dilutions were also carried out in methanol.

### 3.2. Environmental Samples

Detailed descriptions of the collection of samples of desert dusts, marine particulate matter and sediments used for validation of the proposed 3β,5α,6β-steratriol derivatization method can be found elsewhere [39,40,41,42]. Treatment of the whole material of the different samples involved reduction with excess NaBH_4_ in MeOH (25 mL; 30 min) to convert labile hydroperoxides (resulting from oxidation) to their corresponding alcohols, which are more amenable to analysis using GC-EIMS, GC-EIMS/MS and GC-QTOF. Water (25 mL) and KOH (2.8 g) were then added and the resulting mixture saponified by refluxing (2 h). After cooling, the mixture was acidified (HCl, 2 N) to pH 1 and extracted with dichloromethane (DCM; 3 × 20 mL). The combined DCM extracts were dried over anhydrous Na_2_SO_4_, filtered and concentrated via rotary evaporation at 40 °C to give TLEs. All the solvents (pesticide/glass distilled grade) and reagents (Puriss grade) were obtained from Rathburn (Walkerburn, Scotland) and Sigma-Aldrich (Saint Quentin Fallavier, France), respectively. The different TLEs obtained were derivatized as described in the following section.

### 3.3. Trifluoroacetylation Method

In an effort to optimize the derivatization reaction, we tested several parameters, including the nature of the solvent (cyclohexane, tetrahydrofuran (THF), diethyl ether, dichloromethane, ethyl acetate and 1,4-dioxane), reaction temperature (50–100 °C), heating time (1–24 h) and volume of TFAA (25–200 µL). Although this reaction could be also carried out with pentafluoropropionic anhydride, we selected TFAA as the derivatizing reagent since it allowed the formation of fluorinated derivatives with a better yield (>95%). The best reaction efficiency was obtained with the following conditions.

Samples to be derivatized (after evaporation to dryness under a stream of nitrogen at 50 °C) (2–100 ng), internal standard (66 ng), anhydrous THF (200 µL) and TFAA (100 µL) were put in glass vials (4 mL) with PTFE-lined screw caps, and the mixtures were maintained at 68–70 °C in a heating block for 24 h. After evaporation to dryness under a stream of nitrogen at 50 °C, the residues were dissolved in BSTFA to silylate the traces of trifluoroacetic acid formed during the reaction that could damage the GC column employed.

### 3.4. Silylation

3β,5α,6β-steratriols were silylated by dissolving them in 300 µL measures of a mixture of pyridine and BSTFA (2:1, *v*/*v*) and heating to 50 °C for 1 h. After evaporation to dryness under a stream of N_2_ at 50 °C, the derivatized residues were dissolved in BSTFA.

### 3.5. Gas Chromatography-Tandem Electron Ionization Mass Spectrometry (GC-EIMS/MS)

GC-EIMS and GC-EIMS/MS analyses were performed using an Agilent 7890A/7010A tandem quadrupole gas chromatograph system (Agilent Technologies, Les Ulis, France) with a cross-linked 5% phenyl-methylpolysiloxane capillary column (Agilent, Courtaboeuf, Les Ulis, France; HP-5MS ultra inert, 30 m × 0.25 mm, 0.25 µm film thickness). Analyses were performed with an injector operating in pulsed splitless mode (1.7 × 10^5^ Pa for 0.5 min) set at 270 °C. Oven temperature was ramped from 70 °C to 130 °C at 20 °C min^−1^, then to 250° C at 5 °C min^−1^ and then to 300 °C at 3 °C min^−1^. The pressure of the carrier gas (He) was held at 0.76 × 10^5^ Pa until the end of the temperature program. The mass spectrometer conditions were as follows: electron energy, 70 eV; source temperature, 230 °C; quadrupole 1 temperature, 150 °C; quadrupole 2 temperature, 150 °C; collision gas (N_2_) flow, 1.5 mL min^−1^; quench gas (He) flow, 2.25 mL min^−1^; mass range, *m/z* 50–700; cycle time, 313 ms. Steratriol derivatives were quantified in SIM and MRM modes. Target and precursor ions were selected from the most intense and specific fragmentations observed in the electron ionization (EI) mass spectra. Collision-induced dissociation (CID) was optimized using collision energies ranging from 0 to 20 eV. Quantification with Mass Hunter software (Agilent Technologies, Les Ulis, France) involved peak integration and quantitative determination using calibration curves and ratios between areas of triol and internal standard (cholesta-25,26,26,26,27,27,27-d_7_-3β,5α,6β-triol).

ECNI analyses were carried out on the same apparatus with methane as the reagent gas at 50 mA emission current and 195 eV electron energy. During the experiment, the temperature of the source was held at 150 °C and reactant gas flow was 0.5–0.7 mL min^−1^.

### 3.6. Gas Chromatography-EI Quadrupole Time-of-Flight Mass Spectrometry (GC-QTOF)

Accurate mass measurements were carried out in full scan mode using an Agilent 7890B/7200 GC/QTOF system (Agilent Technologies, Les Ulis, France) with a cross-linked 5% phenyl methylpolysiloxane capillary column (Agilent Technologies; HP-5MS Ultra inert, 30 m × 0.25 mm, 0.25 µm film thickness). Analyses were performed with an injector operating in pulsed splitless mode (1.7 × 10^5^ Pa for 0.5 min) set at 270 °C. Oven temperature was ramped from 70 °C to 130 °C at 20 °C min^−1^ and then to 300 °C at 5 °C min^−1^. The pressure of the carrier gas (He) was held at 0.76 × 10^5^ Pa until the end of the temperature program. Instrument temperatures were 300 °C for the transfer line and 230 °C for the ion source. Nitrogen (1.5 mL min^−1^) was used as the collision gas. Accurate mass spectra were recorded across the range of *m/z* 50–700 at 4 GHz with the collision gas opened. The QTOF-MS instrument provided a typical resolution ranging from 8009 to 12,252 from *m/z* 68.9955 to 501.9706. Perfluorotributylamine (PFTBA) was used for daily MS calibration. Compounds were identified by comparing their EI(TOF) mass spectra, accurate masses and retention times against standards.

## 4. Conclusions

In this study, we developed a new method of derivatization of 3β,5α,6β-steratriols (trifluoroacetylation), allowing to considerably improve the quantification of these sterol oxidation products (used as tracers of autoxidation processes) in natural samples. Indeed, the trifluoroacetylated derivatives presented much better chromatographic characteristics (shorter retention times and finer peaks) and were much more stable than the corresponding trimethylsilyl derivatives most often used to quantify 3β,5α,6β-steratriols. This new derivatization method should therefore allow a much more accurate estimation of the oxidation state of sterols in environmental samples. It also allows the separation of diastereoisomeric steratriols, which is not possible with disilylated derivatives. Note that this method allowed the detection of 3β,5α,6β-steratriols derived from the oxidation of diunsaturated sterols that have never been described before in the literature. 

## Figures and Tables

**Figure 1 molecules-28-01547-f001:**
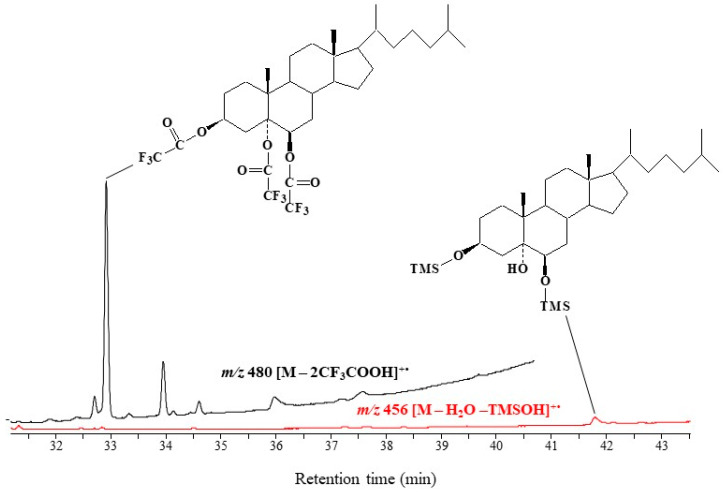
SIM chromatograms of the same amounts (46.5 ng) of trifluoroacetate (black) and trimethylsilyl (red) derivatives of cholesta-3β,5α,6β-triol (• = radical, +• = radical cation).

**Figure 2 molecules-28-01547-f002:**
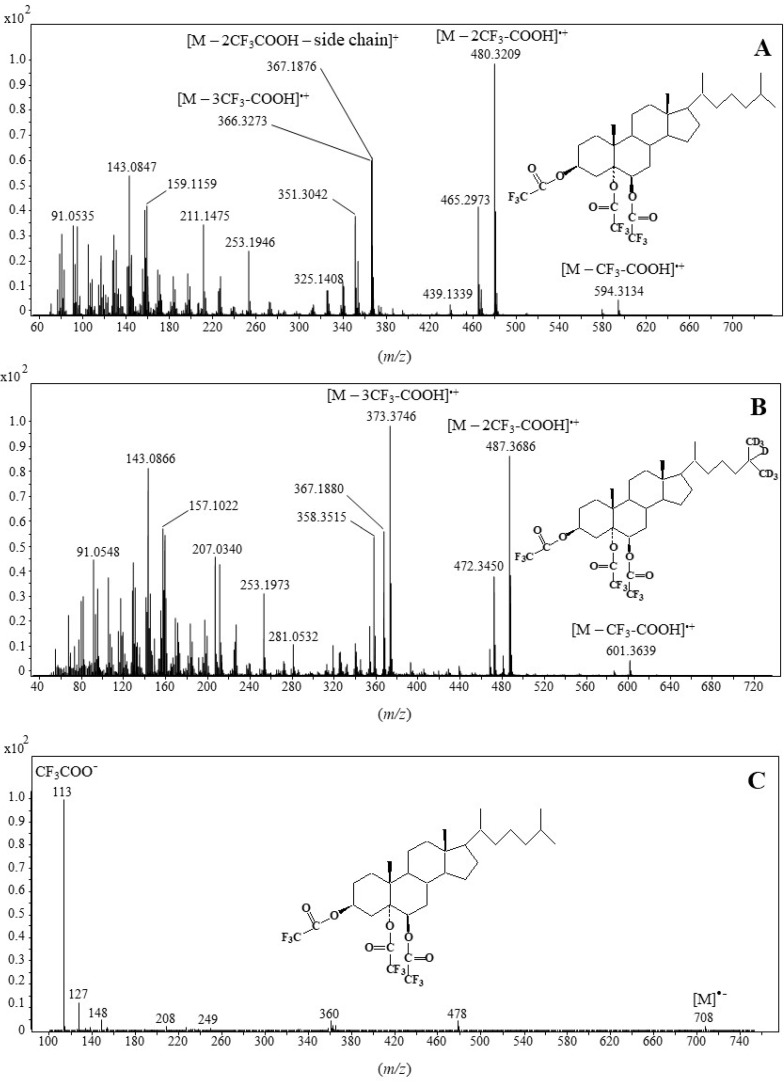
EI(TOF) mass spectra of cholesta-3β,5α,6β-triol (**A**) and cholesta-25,26,26,26,27,27,27-d_7_-3β,5α,6β-triol (**B**) trifluoroacetate derivatives and electron-capture negative ionization (ECNI) mass spectrum of cholesta-3β,5α,6β-triol trifluoroacetate derivative (**C**) (• = radical, •− = radical anion).

**Figure 3 molecules-28-01547-f003:**
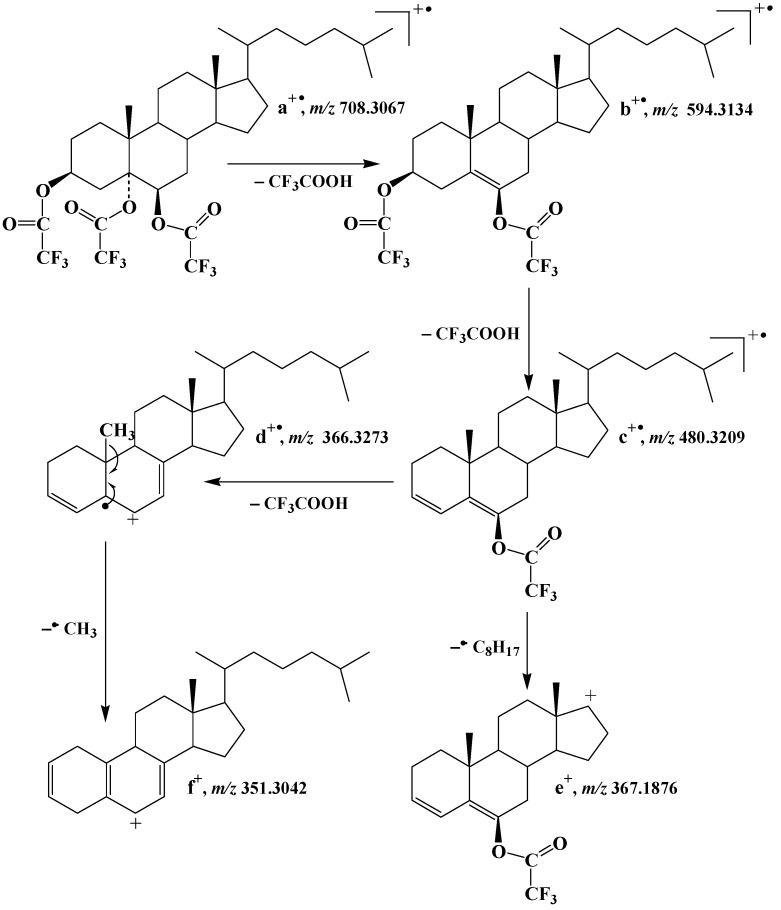
Proposed fragmentation of cholesta-3β,5α,6β-triol trifluoroacetate derivative. Note that another mechanism (not shown) involving initial loss of the 3β-acyl group is also possible (• = radical, +• = radical cation, + = cation, − = loss).

**Figure 4 molecules-28-01547-f004:**
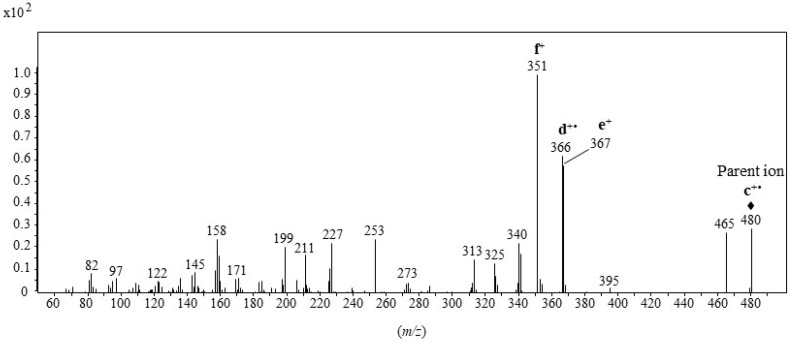
CID mass spectrum of the **c^+•^** ion at *m/z* 480 (collision energy: 14 eV) (+• = radical cation, + = cation, ⧫ = parent ion).

**Figure 5 molecules-28-01547-f005:**
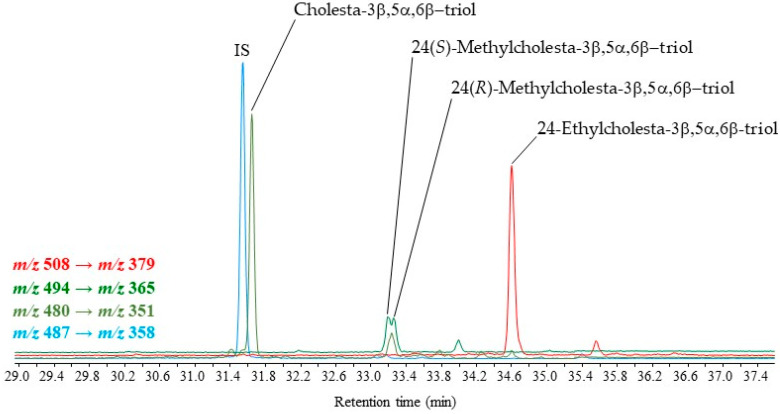
MRM chromatograms showing the presence of 3β,5α,6β-steratriol trifluoroacetate derivatives in a total lipid extract (TLE) of dusts collected in the Negev desert (Israel) (IS = internal standard).

**Figure 6 molecules-28-01547-f006:**
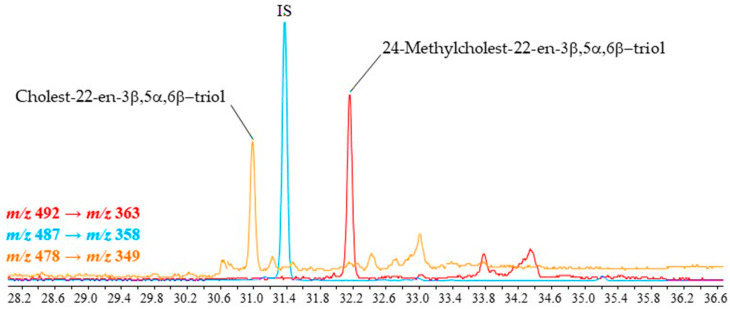
MRM chromatograms showing the presence of unsaturated 3β,5α,6β-steratriol trifluoroacetate derivatives in TLE of suspended particulate matter collected in the Amundsen Sea (Antarctica) (IS = internal standard).

**Figure 7 molecules-28-01547-f007:**
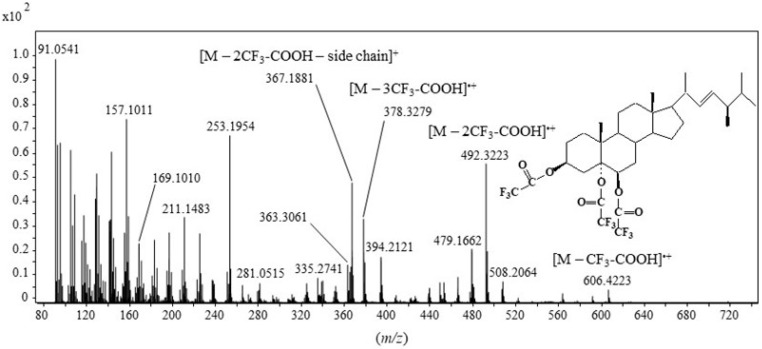
TOF mass spectrum of 24-methylcholest-22*E*-en3β,5α,6β-triol trifluoroacetate derivative. • = radical, •+ = radical cation, + = cation.

**Table 1 molecules-28-01547-t001:** MRM transitions employed for the quantification of 3β,5α,6β-steratriols originating from autoxidation of common Δ^5^-sterols (+• = radical cation, + = cation).

3β,5α,6β-Steratriols	c^+•^ Parent Ion [M—2CF_3_COOH]^+•^	f^+^ Product Ion [M—2CF_3_COOH—CH_3_]^+^	CE * (eV)
Cholesta-3β,5α,6β-triol	480	351	14
Cholest-22-en3β,5α,6β-triol	478	349	14
24-Methylcholest-22-en3β,5α,6β-triol	492	363	14
24-Methylcholest-24(28)-en3β,5α,6β-triol	492	363	14
24-Methylcholesta-3β,5α,6β-triol	494	365	14
24-Ethylcholest-22-en3β,5α,6β-triol	506	377	14
24-Ethylcholesta-3β,5α,6β-triol	508	379	14

* Collision energy.

**Table 2 molecules-28-01547-t002:** Linearity in SIM and MRM modes.

Mode	Concentration Range (ng/mL)	Linear Regression Equation	Coefficient of Determination (R^2^)
SIM			
Ion *m/z* 480	2.3–46.5 ^a^	y = 0.0315x − 0.0202	0.9952
Ion *m/z* 594	2.3–46.5	y = 0.0324x − 0.0029	0.9995
MRM			
*m/z* 480 → *m/z* 351	2.3–46.5	y = 0.0358x − 0.0194	0.9974

^a^ (2.325, 4.65, 9.3, 18.6, 23.25, 46.5).

**Table 3 molecules-28-01547-t003:** Reproducibility in SIM and MRM modes tested using different concentrations of cholesta-3β,5α,6β-triol (46.5, 18.6, 9.3, 2.325 ng) mixed with 66 ng of internal standard.

Mode	Concentration (ng/mL)			Relative Standard Deviation * (%)	Difference between Obtained and Expected Concentration (%)
	Expected	Obtained	*n*		
SIM					
Ion *m/z* 480	46.5	44.7	6	4.4	−3.9
	18.6	18.4	9	8.7	−1.1
	9.3	8.9	8	3.8	−4.3
	2.325	2.618	6	9.9	12.6
Ion *m/z* 594	46.5	44.4	6	5.4	−4.5
	18.6	18.3	9	8.5	−1.6
	9.3	9.2	8	3.0	−1.1
	2.325	2.696	6	4.2	15.9
MRM					
*m/z* 480 → *m/z* 351	46.5	45.6	6	3.1	−1.9
	18.6	18.63	9	4.1	0.2
	9.3	9.0	8	3.7	−3.2
	2.325	2.863	6	4.2	10.2

* 95% confidence.

**Table 4 molecules-28-01547-t004:** Concentrations of saturated 3β,5α,6β-steratriols in several environmental samples, measured after trifluoroacetylation in SIM and MRM modes.

	Cholesta-3β,5α,6β-Triol		24-Methylcholesta- 3β,5α,6β-Triols ^d^		24-Ethylcholesta-3β,5α,6β-Triol	
	SIM *m/z* 480	MRM *m/z* 480 → *m/z* 351	SIM *m/z* 494	MRM *m/z* 494 → *m/z* 365	SIM *m/z* 508	MRM *m/z* 508 → *m/z* 379
Negev loess sample 1 ^a^	0.21	0.21	0.09	0.06	0.24	0.17
Negev loess sample 2 ^a^	0.24	0.19	0.07	0.05	0.19	0.12
Negev loess sample 3 ^a^	0.23	0.22	0.12	0.09	0.35	0.25
Particles Antarctica st 4 ^b^	11.4	11.73	0.80	nd ^c^	1.95	1.25
Particles Antarctica st 13 ^b^	7.11	8.10	0.85	1.00	2.00	1.25
Particles Antarctica st 28 ^b^	6.22	7.40	0.95	0.95	2.00	1.10
Particles Antarctica st 42 ^b^	6.08	6.45	1.00	nd ^c^	1.70	0.90
Particles Antarctica st 46 ^b^	11.73	11.88	1.05	1.00	1.85	1.15
Sediment Baffin Bay st 600 ^a^	20.30	21.05	5.63	4.73	20.27	17.91
Sediment Baffin Bay st 605 ^a^	22.98	23.56	8.67	5.33	37.67	24.11
Sediment Baffin Bay st 615 ^a^	19.32	19.32	4.11	3.47	26.68	17.21
Sediment Baffin Bay st 707 ^a^	45.60	49.28	2.20	2.20	37.60	40.00
Sediment Baffin Bay st 719 ^a^	18.19	17.87	5.00	3.27	24.33	13.5

^a^ ng mg^−1^, ^b^ ng L^−1^, ^c^ Not detected, ^d^ Sum of diastereoisomers.

**Table 5 molecules-28-01547-t005:** Concentrations of unsaturated 3β,5α,6β-steratriols in several environmental samples, measured after trifluoroacetylation in SIM and MRM modes.

	Cholest-22*E*-en-3β,5α,6β-Triol		24-Ethylcholest-22*E*-en-3β,5α,6β-Triol		24-Methylcholest-22*E*-en-3β,5α,6β-Triol	
	SIM *m/z* 506	MRM *m/z* 506 → *m/z* 377	SIM *m/z* 478	MRM *m/z* 478 → *m/z* 349	SIM *m/z* 492	MRM *m/z* 492 → *m/z* 363
Negev loess sample 1 ^a^	- ^d^	0.11	nd ^c^	0.04	nd ^c^	0.02
Negev loess sample 2 ^a^	- ^d^	0.02	nd ^c^	nd ^c^	nd ^c^	nd ^c^
Negev loess sample 3 ^a^	- ^d^	0.01	nd ^c^	nd ^c^	nd ^c^	nd ^c^
Particles Antarctica st 4 ^b^	- ^d^	1.42	1.11	nd ^c^	1.11	1.80
Particles Antarctica st 13 ^b^	- ^d^	1.50	nd ^c^	nd ^c^	nd ^c^	1.40
Particles Antarctica st 28 ^b^	- ^d^	1.60	nd ^c^	nd ^c^	nd ^c^	3.70
Particles Antarctica st 42 ^b^	- ^d^	1.85	nd ^c^	nd ^c^	nd ^c^	2.20
Particles Antarctica st 46 ^b^	- ^d^	1.95	nd ^c^	nd ^c^	nd ^c^	1.70
Sediment Baffin Bay st 600 ^a^	- ^d^	4.09	3.80	1.45	1.64	1.63
Sediment Baffin Bay st 605 ^a^	- ^d^	6.22	4.67	1.78	1.78	1.78
Sediment Baffin Bay st 615 ^a^	- ^d^	2.68	nd ^c^	1.26	1.84	1.84
Sediment Baffin Bay st 707 ^a^	- ^d^	6.20	nd ^c^	3.40	nd ^c^	5.40
Sediment Baffin Bay st 719 ^a^	- ^d^	1.07	nd ^c^	1.07	1.40	1.40

^a^ ng mg^−1^, ^b^ ng L^−1^, ^c^ Not detected,^.d^ Quantification hindered by a strong coelution.

## Data Availability

Not applicable.

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
