# Peer review of "Use of Trifluoro-Acetate Derivatives for GC-MS and GC-MS/MS Quantification of Trace Amounts of Stera-3β,5α,6β-Triols (Tracers of Δ5-Sterol Autoxidation) in Environmental Samples"

_molecules, 2023, doi:10.3390/molecules28041547_

Round 1

Reviewer 1 Report

The manuscript anticipates to present a developed method to improve detection of stera-3ß,5a,6ß-triol (pls explain why plural is used). The study seems to be a valid approach with good results regarding the claimed objective; however, my main concern is the weak discussion of the derivatization reaction and lacking presentation of the achievements in comparison to the reference method, i.e. silylation. Thus, in the beginning the authors fail to summarize the state of the art, i.e. when this derivatization method was first applied to sterols before, what are the drawbacks and why silylation became more popular instead. Likewise, instead of long, descriptive listing of steroid concentrations in several samples, they fail to present their results of method development in sufficient detail in terms of their claims in the abstract, according to which their method provides derivatives with "(i) better stability, (ii) shorter retention times, (iii) better chromatographic properties, and (iv) mass spectra featuring specific ions or transitions that enable a very low limit of detection in selected ion monitoring (SIM) and multiple reaction monitoring (MRM) modes." and provide only exemplary information here. If this is a method paper, more detailed information should be given on the (different?) analyte(s?) and be compared to the reference method. Also, pls make clear if according to their results this is a general claim or only applies to just one compound?

In the following, I also add some minor comments:

line 39: electrospray not electron spray

line 51: typo tailing

line 57: factor of what?

line 69: Figure 1. The area of the silylated derivative seems larger; are both measurements done with the same compound concentration?

Line 102: Figure 3. The authors use the concept of localized charge to formulate the fragmentation mechanism. Would they pls elaborate step one, i.e. charge and radical migration from C-3 O to C-6 O? And, is there any experimental evidence that the middle TFA-residue is cleaved first? After the first cleavage, the concept of localized charge is avoided then; why that? Pls comment.

line 124: milliliter is abbreviated by "mL", replace throughout the manuscript

line 130: table 2. R2 is the coefficient of determination, pls correct

line 133: pls provide a description, possibly in the Mat and Meth section, how the "found" concentration was actually calculated? Did you use the internal standard? You claim that: "... accuracy ... were acceptable over the concentration range" but you show only the two concentration levels in the middle; pls provide the values at the end of the dynamic range as well.

Line 140: did you also use MRM to determine the LOD of the disilylated derivative and which transition?

Line 235: give the exp details of the pulse ("pulsed splitless")

line 241: do you really mean Daltons or better m/z?

line 250: % is not a unit of a gas flow. 40% of what?

Author Response

Answers to the reviewer 1 comments

  • Comment 1 : my main concern is the weak discussion of the derivatization reaction and lacking presentation of the achievements in comparison to the reference method, i.e. silylation. Thus, in the beginning the authors fail to summarize the state of the art, i.e. when this derivatization method was first applied to sterols before, what are the drawbacks and why silylation became more popular instead.

  • Answer : Some text and references where added in the introduction as required (see text lines 43-51).

  • Comment 2 : Likewise, instead of long, descriptive listing of steroid concentrations in several samples, they fail to present their results of method development in sufficient detail in terms of their claims in the abstract, according to which their method provides derivatives with "(i) better stability, (ii) shorter retention times, (iii) better chromatographic properties, and (iv) mass spectra featuring specific ions or transitions that enable a very low limit of detection in selected ion monitoring (SIM) and multiple reaction monitoring (MRM) modes." and provide only exemplary information here. If this is a method paper, more detailed information should be given on the (different?) analyte(s?) and be compared to the reference method. Also, pls make clear if according to their results this is a general claim or only applies to just one compound?

  • Answer : The stability of acyl and trimethylsilyl derivatives is compared lines 82-84. The new figure 1 (comparing now the chromatograms of the same amount of TMS and trifluoroacyl derivatives of cholesta-3β,5a,6b-triol) clearly shows the shorter retention time and the better chromatographic properties of the acyl derivative. It is now clearly indicated in the abstract that the method was validated with cholesta-3β,5a,6b-triol.

  • Comment 3 : line 39: electrospray not electron spray

  •  

  • Comment 4 : Line 51: typo tailing

  •  

  • Comment 5 : line 57: factor of what?

  • The sentence was changed (see lines 63-64).

  • Comment 6 : line 69: Figure 1. The area of the silylated derivative seems larger; are both measurements done with the same compound concentration.

  • Answer : No! But this figure was replaced by another showing the chromatograms of the same amount (25 ng) of trimethylsilyl and trifluoroacyl derivatives of cholesta-3β,5a,6b-triol.

  • Comment 7 : Line 102: Figure 3. The authors use the concept of localized charge to formulate the fragmentation mechanism. Would they pls elaborate step one, i.e. charge and radical migration from C-3 O to C-6 O? And, is there any experimental evidence that the middle TFA-residue is cleaved first? After the first cleavage, the concept of localized charge is avoided then; why that? Pls comment.

  • Answer : To simplify the scheme, only global charges are now given for the losses of the three neutral molecules of trifluoroacetic acid. The strong abundance of the fragment ion c+ at m/z 480 results from its stable enol ester structure. Such a structure can be obtained after the initial loss of the 5a- or 3b- acyl group. It is now indicated in the caption of the figure that the two possibilities exist.

  • Comment 8 : line 124: milliliter is abbreviated by "mL", replace throughout the manuscript.

  • Answer : Done.

  • Comment 9 : line 130: table 2. R2 is the coefficient of determination, pls correct

  • Answer : Done.

  • Comment 10 : ine 133: pls provide a description, possibly in the Mat and Meth section, how the "found" concentration was actually calculated? Did you use the internal standard? You claim that: "... accuracy ... were acceptable over the concentration range" but you show only the two concentration levels in the middle; pls provide the values at the end of the dynamic range as well.

  • Answer : Details of the quantification were added in material and method part (see lines 246-249). The table 3 was changed and it contains now the values of the end of the dynamic range.

  • Comment 11 : Line 140: did you also use MRM to determine the LOD of the disilylated derivative and which transition?

  • Answer : No the comparison was carried out in SIM mode with the target ion m/z 456 corresponding to [M - TMSOH – H2O]+•. This is now indicated in the text (see lines 145-146).

  • Comment 12 : Line 235: give the exp details of the pulse ("pulsed splitless")

  • Answer : The details were added in the text.

  • Comment 13 : line 241: do you really mean Daltons or better m/z?

  • Answer : Daltons was replaced by m/z.

  • Comment 14 : line 250: % is not a unit of a gas flow. 40% of what?

  • Answer : The flow of reactant gas is now indicated.

Reviewer 2 Report

the manuscript is well written. The data presented by the authors support the discussion and conclusion.

Author Response

The English language was checked.

Reviewer 3 Report

You have presented an original and highly scientific study.

Author Response

The English language was checked.

Round 2

Reviewer 1 Report

earlier Comment 1: The authors did not provide a summary of state-of-the art for acetylation of steroids with TFA; for example, in Line 60 this could be well added. Thus, for acetylation of steroids, most of the literature reports on the use of acetic acid anhydride (eg: Angelis et al. Drug Test. Analysis 2012, 4, 923–927), but for analysis, TFA derivatization of steroids is also common (Detection of anabolic steroids by GC/SIM/MS with trifluoroacetylation in equine plasma and urine, Choi, Man, Analytical Letters, 32(7):1313-1322, 1999; M. Harnik, E. Hürzeler, E.V. Jensen Tetrahedron Volume 23, Issue 1, 1967, Pages 335-340 Acetylation and trifluoroacetylation of steroids at carbon 16). In what the described method is different to already-published methods, improving the outcome of their methods for steroid analysis?

earlier Comment 2: It is not clear from the manuscript text, if the authors tested the stability of their derivatives, which is claimed in the abstract. Pls specify as appropriate. If they did not test stability themselves, this information should be removed from the abstract and just discussed in the text.

earlier Comment 9: R2 is defined as coefficient of determination (correct in the table header); if the authors really want to provide the correlation coefficient though, they should list R.

earlier Comment 10: How much internal standard was spiked? It might also enhance the comprehension to add some more description to table 3 legend; I guess the “added concentration” relates to the used cholestatriol in standard solution? Or was it spiked to any matrix?

Earlier comment 11: you optimized the conditions for acetylation but not for silylation; as such, the ratio of pyridine to BSTFA possibly might be better chosen reverse, i.e. 1:2? The SIM LOD of acetylation is actually better by factor 2; do you think that with an optimized silylation one could match the sensitivity of acetylation? Pls comment.

Legend Figure 2: TOF is a mass analyzer but this figure seems to compare ionization techniques, it should be added that A and B are EI spectra

Author Response

Answers to the reviewer comments

earlier Comment 1: The authors did not provide a summary of state-of-the art for acetylation of steroids with TFA; for example, in Line 60 this could be well added. Thus, for acetylation of steroids, most of the literature reports on the use of acetic acid anhydride (eg: Angelis et al. Drug Test. Analysis 2012, 4, 923–927), but for analysis, TFA derivatization of steroids is also common (Detection of anabolic steroids by GC/SIM/MS with trifluoroacetylation in equine plasma and urine, Choi, Man, Analytical Letters, 32(7):1313-1322, 1999; M. Harnik, E. Hürzeler, E.V. Jensen Tetrahedron Volume 23, Issue 1, 1967, Pages 335-340 Acetylation and trifluoroacetylation of steroids at carbon 16). In what the described method is different to already-published methods, improving the outcome of their methods for steroid analysis?

Answer : As required, some text concerning acylation and perfluoroacylation of steroids was added in the introduction (see lines 60-63).

earlier Comment 2: It is not clear from the manuscript text, if the authors tested the stability of their derivatives, which is claimed in the abstract. Pls specify as appropriate. If they did not test stability themselves, this information should be removed from the abstract and just discussed in the text.

Answer : It was previously indicated in the text (lines 87-88 in the new version) that the stability of the trifluoroacetate derivative of cholesta-3β,5a,6b-triol was tested and compared to this of the disilylated derivative. The trifluoroacetate derivative could be stored at 4°C for several months without significant alteration, while the silylated derivative was hydrolyzed in few days.

earlier Comment 9: R2 is defined as coefficient of determination (correct in the table header); if the authors really want to provide the correlation coefficient though, they should list R.

Answer : The changes were done.

earlier Comment 10: How much internal standard was spiked? It might also enhance the comprehension to add some more description to table 3 legend; I guess the “added concentration” relates to the used cholestatriol in standard solution? Or was it spiked to any matrix?

Answer : The amount of IS added is now indicated in the material and method (see line 229). Some indications were also added in the caption of Table 3. Details concerning the preparation of standard solutions were also given in the experimental part (see lines 216-218). Added concentrations related effectively to standard solutions of cholestatriol.

Earlier comment 11: you optimized the conditions for acetylation but not for silylation; as such, the ratio of pyridine to BSTFA possibly might be better chosen reverse, i.e. 1:2? The SIM LOD of acetylation is actually better by factor 2; do you think that with an optimized silylation one could match the sensitivity of acetylation? Pls comment.

Answer : SIM LOD of the disilylated derivative is 620 pg (0.62 ng) while this of the trifluoroacetate derivative is 25.8 pg (see text lines 148-150), acetylation is thus better by a factor 24 and not 2 than silylation. There was an error in the Figure 1 where the two chromatograms did not really correspond to the same amount of derivatives. This figure was replaced by another one showing SIM chromatograms of the same amount (46.5 ng) of the two derivatives. Only the trisilylated derivative could challenge the sensitivity of the trifluoroacetate derivative. As it is indicated in the introduction, the three hydroxyl groups of steratriols can only be silylated with a mixture of DMSO and BSTFA. Unfortunately, this derivatisation method is not complete (yield < 60%) and needs the elimination of DMSO (after addition of water and solvent extraction) and resilylation of the labile positions 3 and 6 before injection. It is thus inapplicable for trace quantification. 

Legend Figure 2: TOF is a mass analyzer but this figure seems to compare ionization techniques, it should be added that A and B are EI spectra

Answer : TOF was replaced par EI(TOF).